

# Growth parameter *k* and location affect body size responses to spatial protection by exploited rockfishes

Madeleine McGreer[1,*], Alejandro Frid[1,2,*], Tristan Blaine[1], Sandie Hankewich[3], Ernest Mason[3], Mike Reid[4] and Hannah Kobluk[1]

[1] Central Coast Indigenous Resource Alliance, Campbell River, British Columbia, Canada
[2] School of Environmental Studies, University of Victoria, Victoria, British Columbia, Canada
[3] Kitasoo/Xai'xais Fisheries, Klemtu, British Columbia, Canada
[4] Heiltsuk Integrated Resource Management Department, Bella Bella, British Columbia, Canada
[*] These authors contributed equally to this work.

Corresponding author
Alejandro Frid, alejfrid@gmail.com

## ABSTRACT

For many fish taxa, trophic position and relative fecundity increase with body size, yet fisheries remove the largest individuals, altering food webs and reducing population productivity. Marine reserves and other forms of spatial protection can help mitigate this problem, but the effectiveness of these management tools may vary interspecifically and spatially. Using visual survey data collected on the Central Coast of British Columbia, for 12 species of exploited rockfish we found that body size responses to spatial fishery closures depended on interspecific variation in growth parameter *k* (the rate at which the asymptotic body size is approached) and on location. For two closures, relative body sizes were larger at protected than at adjacent fished sites, and these differences were greater for species with lower *k* values. Reduced fishery mortality likely drove these results, as an unfished species did not respond to spatial protection. For three closures, however, body sizes did not differ between protected and adjacent fished sites, and for another closure species with higher *k* values were larger at fished than at protected sites while species with lower *k* values had similar sizes in both treatments. Variation in the age of closures is unlikely to have influenced results, as most data were collected when closures were 13 to 15-years-old. Rather, the lack of larger fish inside four of six spatial fishery closures potentially reflects a combination of smaller size of the area protected, poor fisher compliance, and lower oceanographic productivity. Interspecific differences in movement behavior did not affect body size responses to spatial protection. To improve understanding, additional research should be conducted at deeper depths encompassing the distribution of older, larger fish. Our study—which was conceptualized and executed by an alliance of Indigenous peoples seeking to restore rockfishes—illustrates how life history and behavioral theory provide a useful lens for framing and interpreting species differences in responses to spatial protection.

## INTRODUCTION

For centuries, many Indigenous cultures have recognized that protecting parts of the ocean from exploitation mitigates human impacts on biodiversity and contributes to fishery sustainability (*Johannes, 1978*; *Jones, Rigg & Lee, 2010*; *Ban et al., 2018*). Consistent with the knowledge and practices of these cultures, there is growing evidence that marine reserves or other forms of spatial protection promote recovery from exploitation (*Baskett & Barnett, 2015*), and that spillover of adults and larvae from protected areas can enhance adjacent fisheries (*Di Lorenzo, Claudet & Guidetti, 2016*; *Baetscher et al., 2019*).

The benefits of spatial protection often are quantified as increases in the density, biomass, and body sizes of exploited species (*Baskett & Barnett, 2015*). Understanding factors affecting body size distributions is particularly important because fisheries remove the largest individuals, often at great ecological cost (*Strong & Frank, 2010*; *Hixon, Johnson & Sogard, 2014*). Larger individuals tend to occupy higher trophic positions (e.g., *Trebilco et al., 2016*; *Olson et al., 2020*), and their removal may alter food web structure (*Shackell et al., 2010*; *Zgliczynski & Sandin, 2017*). Further, for many fish taxa the relationship between fecundity and body size is hyperallometric (a power function with exponents > 1), such that larger females contribute disproportionately more offspring (per unit of body size) than smaller females (*Dick et al., 2017*; *Barneche et al., 2018*). Size differences between protected and fished areas, therefore, can signal the extent to which spatial protection promotes species recoveries and restores food webs.

The benefits of spatial protection, however, can differ across species, and understanding this variation may help predict reserve performance (*Jennings, 2000*; *Claudet et al., 2010*; *Kaplan et al., 2019*). Species with longer lifespans often have slower growth rates, later maturity, and other traits which increase their vulnerability to exploitation and reduce their recovery rates during fishery reprieves (*Jennings, 2000*). Consequently, the benefits of spatial protection may require more time to manifest in longer-lived species (*Starr et al., 2015*; *Kaplan et al., 2019*).

Variation in parameter *k* of the von Bertalanffy growth function, which depicts the rate at which the asymptotic body size is approached, should influence the time required by different species to restore a larger size structure after the implementation of spatial protection (*Jennings, 2000*; *Kaplan et al., 2019*). Additionally, species characterized by smaller movements and greater site fidelity are more likely to remain within a protected area (*Hannah & Rankin, 2011*) and may benefit more from spatial protection than more mobile species (*Kramer & Chapman, 1999*; *Moffitt et al., 2009*). These predictions, however, do not apply to unfished species, for which environmental variation is a primary driver of population characteristics (*Claudet et al., 2010*; *Caselle et al., 2015*).

Rockfish (*Sebastes* spp.), a genus of marine fish with diverse life history traits, allow tests of these ideas. In the northeast Pacific, maximum lifespans range across rockfish species from two decades to two centuries, and interspecific variation in body size and growth parameters are similarly wide (*Love, Yoklavich & Thorsteinson, 2002*). While most rockfishes associate with rocky reefs, their behavior ranges from sedentary to very mobile and from demersal

to benthopelagic. Most rockfishes are exploited yet some small planktivores are unfished (*Love, Yoklavich & Thorsteinson, 2002*).

In British Columbia, Canada, commercial and recreational sectors overfished rockfishes during the latter part of the 20th century, diminishing their abundance and body sizes (*Yamanaka & Logan, 2010*; *Eckert et al., 2017*). During the 2000s federal legislators implemented more conservative management and established a network of spatial fishery closures known as Rockfish Conservation Areas (RCAs) where commercial and recreational fishers cannot use bottom trawls, groundlines, or hook-and-line gear (*Yamanaka & Logan, 2010*; Appendix S1). For some species, however, declines in size and age structure have continued and their biomass remains depressed (*McGreer & Frid, 2017*) and references within). RCAs can help reverse these trends, yet prior studies in British Columbia, conducted only 5–7 years after RCA implementation, found no evidence of larger or more abundant rockfishes inside protected sites (*Haggarty, Shurin & Yamanaka, 2016*; *Olson, Trebilco & Salomon, 2019*). As time since RCA implementation progresses, size and biomass increases are more likely to be detected (*Starr et al., 2015*; *Keller et al., 2019*), particularly for shorter-lived species with greater $k$ values (*Kaplan et al., 2019*).

Analyses of RCA benefits must account for variation in habitat characteristics which, independently of exploitation, affect the distribution and sizes of rockfish. For some demersal species, relative abundance may increase with topographic structural complexity and larger or older individuals tend to use greater depths than smaller or younger individuals (*Love, Yoklavich & Thorsteinson, 2002*). Benthopelagic species, however, have a wider range of vertical movements into the water column (*Love, Yoklavich & Thorsteinson, 2002*; *Hannah & Rankin, 2011*) and their relationships to depth or topographic structural complexity might be more variable.

Canada is among the countries where Indigenous people are using their traditional knowledge and science to improve marine conservation (*Jones, Rigg & Lee, 2010*; *Ban et al., 2018*; *Ban, Wilson & Neasloss, 2020*). Since 2013, the Central Coast Indigenous Resource Alliance (CCIRA)—comprised of the Wuikinuxv, Nuxalk, Heiltsuk and Kitasoo/Xai'xais First Nations—has been surveying rockfish populations and their habitats (*Frid et al., 2018*) inside and outside RCAs of British Columbia's Central Coast (Fig. 1; Appendix S2). Data collected to date represent RCAs that were 8 to 15-years-old and encompass the earliest stage expected for the benefits of spatial protection to manifest for rockfishes (*Keller et al., 2019*; *Kaplan et al., 2019*).

Within this temporal context, we used CCIRA's visual survey data to test the hypothesis that body size responses to spatial protection vary according to species life history traits, movement behavior, and susceptibility to fishing, while controlling for depth and topographic structural complexity. We predicted that the body sizes of exploited species would be larger inside RCAs than in adjacent fished areas, but the strength of this effect would decrease for species which require more time to reach their asymptotic body size (i.e., have lower $k$ values) (*Jennings, 2000*; *Kaplan et al., 2019*) or that are more mobile (*Kramer & Chapman, 1999*; *Moffitt et al., 2009*). We also predicted that the body sizes of Puget Sound rockfish (*S. emphaeus*), a small planktivore unlikely to be caught by fishing

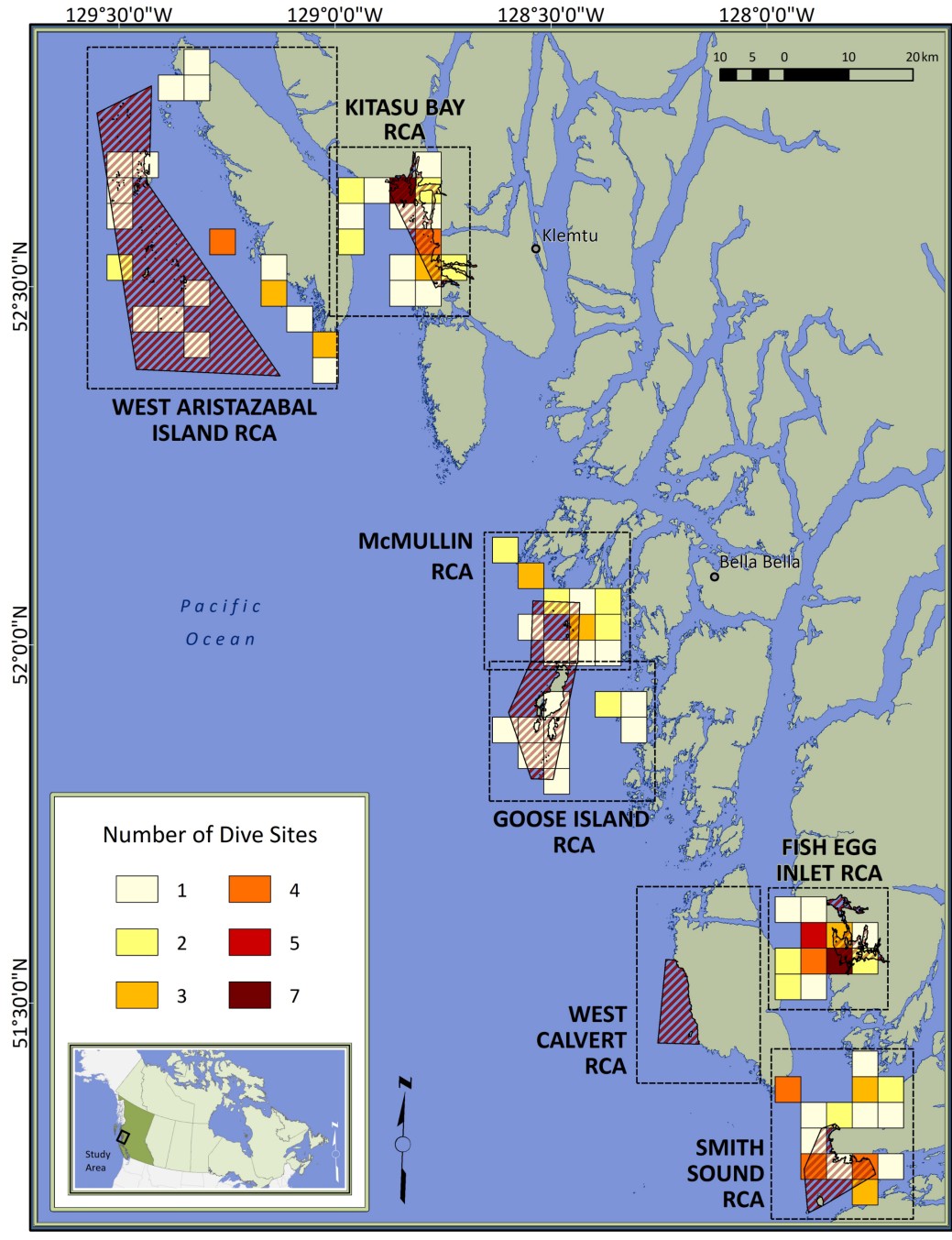

**Figure 1** **Map of the study area.** Red diagonal lines represent Rockfish Conservation Areas (RCA: protected treatments); dashed boxes encompass control sites paired with each RCA; colored squares depict dive locations aggregated within 16-km² grid cells; inset locates the study area in Canada. Individual sites are not identified to protect spatial data considered to be sensitive by First Nations.

gear (*Love, Yoklavich & Thorsteinson, 2002*), would not differ between protected and fished sites (*Claudet et al., 2010*; *Caselle et al., 2015*).

In addition to testing *a priori* predictions, we examined differences in conservation effectiveness between RCA locations. Though exploratory, this aspect of our research was motivated by the need to understand whether some RCAs are performing poorly and the potential ways to improve their management or design (*Haggarty, Martell & Shurin, 2016*; *DFO, 2019a*; *DFO, 2019b*).

## MATERIALS & METHODS

### Ethics statement

Data collection was observational and did not require permits from federal agencies. The Wuikinuxv, Kitasoo/Xai'xais, Heiltsuk and Nuxalk First Nations hold Indigenous rights to their own territories, where all data were collected. Scientific staff conducting field research—who are members of these Nations or work directly for them—had the approvals required by Indigenous rights holders.

### Surveys by SCUBA divers

During March–April or July–October of 2013 and 2015–2019, we sampled rocky reefs, which were located through local Indigenous knowledge or a bathymetric model (*Haggarty & Yamanaka, 2018*). Within the constraints of weather, we attempted to balance sample sizes between paired sites inside RCAs (protected treatment) or outside the same RCA but $\leq$10 km from its nearest boundary (control treatment) (Fig. 1).

SCUBA divers collected visual data on rockfish and their habitats along depth contours within transects that were 30 m-long by 4 m-wide by 4 m-high (480 m$^3$). Dives generally included three transects stratified by depth: 25–35 m, 15–18 m, and $\leq$6 m. Observations included estimates of topographic structural complexity (variation in vertical relief, size, and abundance of rocky structures: Appendix S3), relative density by species (count/480 m$^3$), and total lengths of individual fish. Video was taken during most transects to corroborate habitat descriptions and species identifications.

Total lengths were estimated visually with the aid of a 1-metre-long ruler attached to a pole. Early in their training, the 3 divers who collected these data tested their accuracy measuring fish models attached to a line suspended in the water column. Comparisons of estimated and actual sizes showed low measurement errors that did not vary with model size, and relatively low variation between divers (Appendix S4).

Most fish were encountered alone or in small groups (<20 fish) and sized individually. Benthopelagic species, however, sometimes formed larger schools. In these cases, divers sized fish in clusters, recording a length and a multiplier for the corresponding fish count. This approach allowed divers to avoid long pauses, reducing their influence on attraction or avoidance behaviors that bias counts (*Emslie et al., 2018*). A tradeoff is that size estimates that may have spanned 2–3 cm (had fish been sized individually) were collapsed into a single measurement inflating peaks in the frequency of that measurement (see *Analyses*). For further details on dive surveys, see *Frid et al. (2018)*.

## Species characteristics

We obtained literature values for maximum lifespan, maximum total length, total length at 50% maturity, and von Bertalanffy growth parameter $k$ (Table 1). The latter two parameters vary with latitude (*Love, Yoklavich & Thorsteinson, 2002*), so we prioritized estimates from British Columbia (*Anderson, Keppel & Edwards, 2019*); if unavailable, we used estimates from Oregon and Washington. Species lacking sufficient observations or key information were not analyzed (Appendix S5). The exception was Tiger rockfish (*S. nigrocinctus*), a long-lived species for which we had a large sample size but which lacked data on growth rates; $k$ for this species was estimated as a function of maximum lifespan (Appendices S6, S7).

We used the literature or consulted three experts to class species behaviors according to a combination of relative mobility (high or low) and habitat preference (benthopelagic or demersal) (Table 1; Appendix S8). No species qualified as high mobility-demersal and the one species classed as low mobility-benthopelagic had insufficient observations (Appendix S5). Analyses, therefore, included only 2 behavioral categories: high mobility-benthopelagic and low mobility-demersal (Table 1).

Species were classed as "exploited" or "unfished" based on fishery data compiled by *Anderson, Keppel & Edwards (2019)* or descriptions by *Love, Yoklavich & Thorsteinson (2002)*. Only Puget Sound rockfish qualified as unfished. With the caveat that Puget Sound rockfish have the smallest maximum size in our sample, and therefore shows less body size variation than other rockfishes, we assumed that size distributions for this species are driven primarily by environmental variability, and therefore are an adequate control for the effects of exploitation and protection. Should this assumption be wrong, body size increases under spatial protection are likely to be detected, as Puget Sound rockfish have the highest $k$ value in our sample (Table 1).

## Analyses

We limited analyses to individuals with total lengths of $\geq 10$ cm. Smaller individuals are unlikely to be caught by fishing gear and their abundance is influenced by environmental variability (*Markel, Lotterhos & Robinson, 2017*) outside the scope of analyses.

Total lengths were standardized as length anomalies (LA) which, pooling data from all locations and all years, were calculated for each species as:

$$LA_s = \frac{Ls_i - Ls_{\bar{x}}}{Ls_{std}}$$

where $Ls_i$ is the observed length for individual $i$ belonging to species $s$ and $Ls_{\bar{x}}$ and $Ls_{std}$ are, respectively, the mean and standard deviations of all lengths observed for species $s$. The use of length anomalies, rather than actual lengths, allowed us to include multiple species that differ in maximum body size into a single comprehensive analysis that tested for the effects of species traits on responses to spatial protection. This approach also increased sample sizes, allowing us to include all predictors of interest without overfitting the model.

Length anomaly was the response variable in two generalized least squares models (GLSMs) implemented in R (*Pinheiro et al., 2020*). The first included all exploited species. The predictors were RCA treatment (protected or control) and its two-way interaction

**Table 1 Rockfish species analysed, and their biological characteristics.** Data are from *Love, Yoklavich & Thorsteinson (2002)*, unless noted by superscripts. TL = Total Length. Relative mobility classes are: low = strong site fidelity to small reefs, high = species may or may not exhibit periods of site fidelity to small reefs but also conducts large movements (Appendix S8). For growth parameter k and length at maturity, estimates are for females (F), males (M), and unspecified sex (U). Areas where estimates were obtained are British Columbia (BC), Oregon (OR), Washington (WA).

| *Sebastes* species | Common name | Max. lifespan (yrs) | Max. TL (cm) | Growth parameter *k* | TL at 50% maturity (cm) | Habitat preference | Movement class |
|---|---|---|---|---|---|---|---|
| [*]*S emphaeus* | Puget Sound rockfish | 22 | 18.3 | [*]F = 0.535, WA [*]M = 0.704, WA | [*]F = 12.2, WA | Benthopelagic | [*] |
| *S. melanops* | Black rockfish | 50 | 60 | [1]F = 0.171, WA [1]M = 0.201, WA | [2]F = 41.0, BC [2]M = 42.0, BC | Benthopelagic | High |
| *S. paucispinis* | Bocaccio | 50 | 98 | [2]F = 0.170, BC [2]M = 0.210, BC | [2]F = 52.0, BC [2]M = 48.5, BC | Benthopelagic | High |
| *S. caurinus* | Copper rockfish | 50 | 66 | [2]F = 0.090, BC [2]M = 0.110, BC | [2]F = 25.8, BC [2]M = 26.9, BC | Demersal | Low |
| *S. miniatus* | Vermilion rockfish | 60 | 76 | [3]F = 0.075, OR [3]M = 0.114, OR | [3]F = 39.4, OR | Demersal | Low |
| *S. entomelas* | Widow rockfish | 60 | 53 | [2]F = 0.160, BC [2]M = 0.180, BC | [2]F = 41.3, BC [2]M = 39.9, BC | Benthopelagic | High |
| *S. flavidus* | Yellowtail rockfish | 64 | 66 | [2]F = 0.140, BC [2]M = 0.200, BC | [2]F = 40.9, BC [2]M = 37.6, BC | Benthopelagic | High |
| *S. nebulosus* | China rockfish | 79 | 41 | [4]U=0.147, WA | [2]F = 23.5, BC [2]M = 21.4, BC | Demersal | Low |
| *S. brevispinis* | Silvergray rockfish | 82 | 71 | [2]F = 0.100, BC [2]M = 0.130, BC | [1]F = 43.1, BC [1]M = 42.7, BC | Benthopelagic | High |
| *S. pinniger* | Canary rockfish | 84 | 76 | [2]F = 0.130, BC [2]M = 0.130, BC | [2]F = 42.9, BC [2]M = 40.2, BC | Benthopelagic | High |
| *S. maliger* | Quillback rockfish | 95 | 50 | [2]F = 0.100, BC [2]M = 0.120, BC | [2]F = 27.9, BC [2]M = 29.1, BC | Demersal | Low |
| *S. nigrocinctus* | Tiger rockfish | 116 | 61 | [**] | [1]M = 20.9, BC | Demersal | Low |
| *S. ruberrimus* | Yelloweye rockfish | 121 | 91 | [2]F = 0.040, BC [2]M = 0.050, BC | [1]F = 41.3, BC [1]M = 47.2, BC | Demersal | Low |

**Notes.**

[*]Puget Sound rockfish, a small planktivore, is the only "unfished" species analyzed (*Love, Yoklavich & Thorsteinson, 2002*); growth and maturity are from *Beckmann et al. (1998)*; movement data were unavailable but no required.

[1]Post-2000 estimate of *Cope et al. (2016)*.

[2]*Anderson, Keppel & Edwards (2019)*.

[3]*Hannah & Kautzi (2012)*.

[4]Northern assessment model of *Dick et al. (2016)*.

[**]For tiger rockfish, *k* was derived from regression analyses for the remainder of species (Appendices S6, S7).

with location (i.e., name of RCA associated with protected and control treatments), *k* and its interaction with RCA treatment, behavioral class and its interaction with RCA treatment, topographic structural complexity and its interaction with behavioral class, and depth and its interaction with behavioral class. Parameter *k* values (Table 1) were averaged for males and females, which divers cannot distinguish visually. Because each site contained multiple transects, a Gaussian correlation structure derived from the latitude and longitude of each sampling site (projected into the Albers coordinate system) accounted for the spatial autocorrelation of residuals (*Pinheiro & Bates, 2000*). The second GLSM was specific to the unfished Puget Sound rockfish; its predictors were depth, topographic

structural complexity, and RCA treatment and its interaction with location. This analysis also included the Gaussian correlation structure derived from the coordinates of each sampling site.

Although surveys spanned 6 years, sampling across RCA age-location-treatment combinations was unbalanced (Appendix S9), precluding GLSMs from testing RCA age by treatment interactions. Consequently, analyses pooled data for RCA ages of 8–15 years, with ages 13–15 being best represented (Appendices S9, S10).

To reduce skew in size frequency distributions, multipliers assigned to similarly sized schooling fish were truncated to the 95th percentile of their distribution and incorporated into GLSMs as weights. Competing models were compared with AICc model selection procedures (*Burnham & Anderson, 2002*). We expected the independent effects of topographic structural complexity and depth to affect responses (*Love, Yoklavich & Thorsteinson, 2002*); these predictors did not undergo model selection, reducing the number of competing models. Results are presented in terms of the best GLSMs. Quantile–quantile plots, residuals vs fitted plots, and correlation values between variables were examined to verify model assumptions.

## RESULTS

Of 13,224 observations for exploited species (Appendix S11), 73% were from high mobility-benthopelagic species, and 27% from low mobility-demersal species. The proportion of individuals that had reached or exceeded length at 50% maturity was 47% for low mobility-demersal species and 5% for high mobility-benthopelagic species (Appendix S12).

The effects of spatial protection on the body sizes of exploited species depended on growth parameter $k$ and RCA location, but not on behavioral class (Fig. 2; Appendices S13A, S14A). At two RCAs (West Aristazabal, Goose Island), body sizes were larger at protected than control sites, but these differences were greater for slower- than for faster-growing species (Fig. 3). At three RCAs (Fish Egg Inlet, McMullin Group, Smith Sound), body sizes did not differ between protected and control sites, regardless of $k$ values. At the remaining RCA (Kitasu Bay), species with greater $k$ values were larger at control than at protected sites while species with lower $k$ values had similar sizes in both treatments (Fig. 3). Independently of RCA treatment, the body sizes of high mobility-benthopelagic species increased with depth and decreased with greater topographic structural complexity. In contrast, the body sizes of low mobility-demersal species increased with both depth and topographic structural complexity (Fig. 4).

For the unfished Puget sound rockfish, body sizes increased with depth and topographic structural complexity but did not differ between RCA treatments or locations (Fig. 2B, Appendices S13B, S14B, S15).

## DISCUSSION

We predicted that the body sizes of exploited species would be larger inside RCAs than in adjacent fished areas, but the strength of this effect would decrease for species which require more time to reach their asymptotic body size (i.e., have lower values for growth

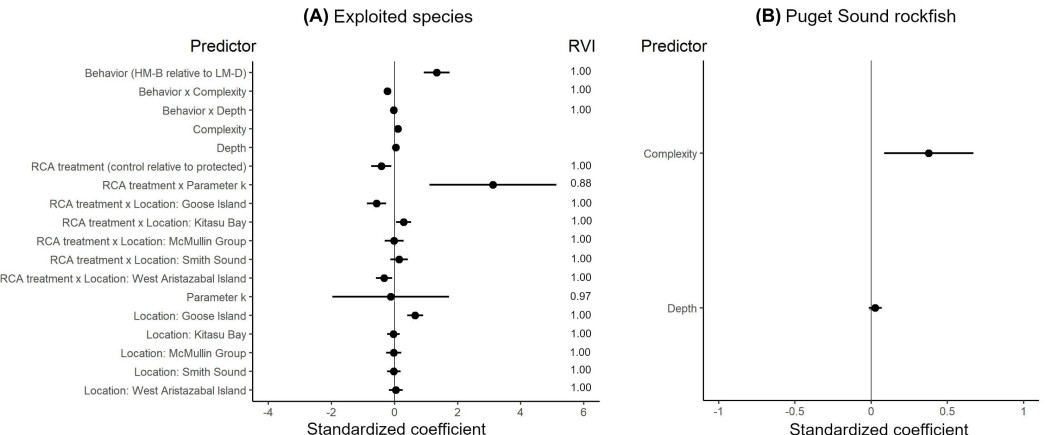

**Figure 2** **Best generalized least squares models describing body size responses to predictors for (A) exploited species and (B) Puget Sound rockfish.** Circles represent standardized coefficients with 95% confidence intervals (horizontal lines). Relative variable importance (RVI) values are provided for predictors that underwent AIC model selection procedures (Appendices S13, S14). Fish Egg Inlet is the baseline for RCA treatment and location. "Complexity" refers to topographic structural complexity. Behaviors are high mobility-benthopelagic (HM-B) or low mobility-demersal (LM-D). Results for exploited species are based on $k = 0.059$ for Tiger rockfish (Appendix S7); results are very similar if the alternative value $k = 0.070$ is used (Appendix S16).

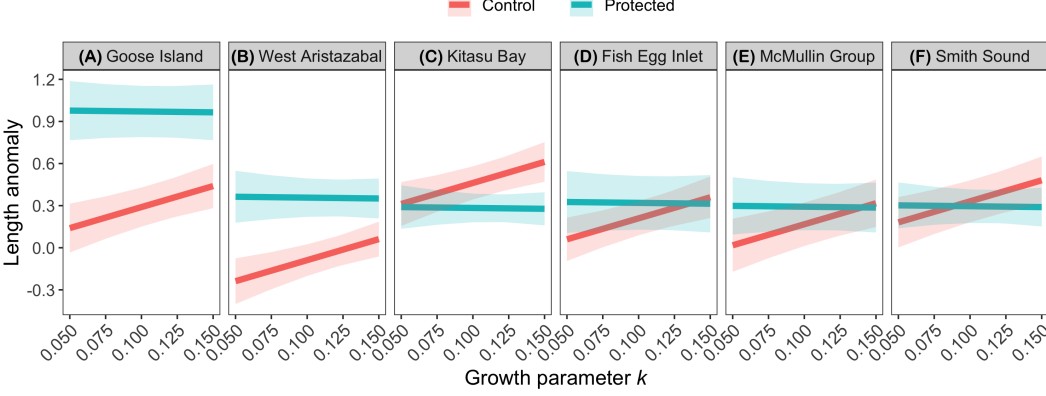

**Figure 3** **Length anomalies of exploited species in relation to parameter *k*, location, and RCA treatment, as estimated by the best generalized least squares model (Fig. 2A).** Estimates are for demersal species at 30 m depth and structural complexity = 3 (patterns remain constant under other conditions). Bands represent 95% confidence intervals. Bands are the 95% confidence interval. For descriptive plots of corresponding raw data, see Appendix S17A.

parameter *k*) or that are more mobile. Our results were inconsistent with these predictions. At two RCAs (Goose Island, West Aristazabal) the body sizes of exploited rockfishes were larger at protected than at adjacent fished sites, but these differences *increased* with lower *k* values. One potential explanation for this result is that body size differences between protected and exploited sites could be influenced by an interaction between parameter *k* and the intensity of exploitation (see *Jennings, 2000*; *Kaplan et al., 2019*); exploitation rates

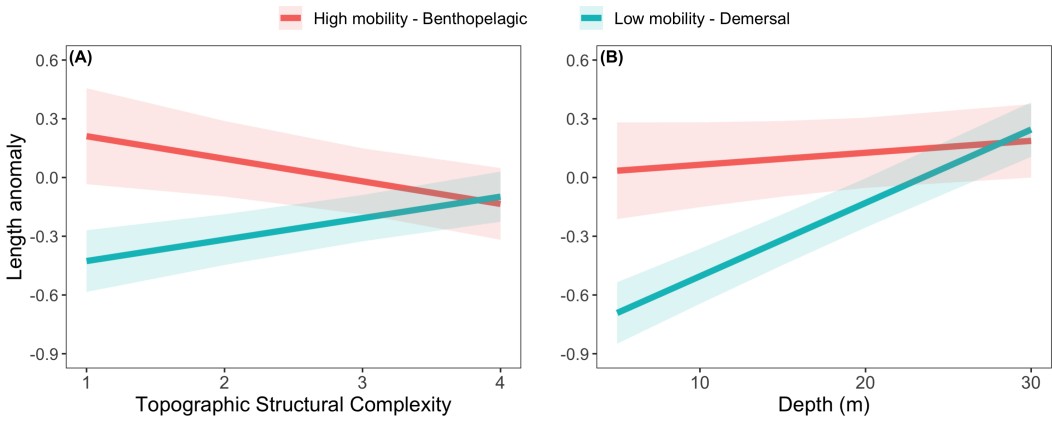

**Figure 4** Length anomalies of exploited rockfishes in relation to (A) topographic structural complexity (at a depth of 15 m) and (B) depth (at complexity = 2), as estimated by the best generalized least square model (Fig. 2A). These estimates are for protected sites in the West Aristazabal RCA and $k = 0.12$ (patterns remain constant under other conditions). Bands are the 95% confidence interval. For descriptive plots of corresponding raw data, see Appendix S17B.

at control sites may have not been high enough to generate strong RCA effects for species with high $k$ values. Also, most rockfishes shift ontogenetically to deeper depths (*Love, Yoklavich & Thorsteinson, 2002*), and the depths that we sampled ($\leq$35 m) encompassed primarily immature fish. Data collected at deeper depths via remotely operated vehicles (e.g., *Haggarty, Shurin & Yamanaka, 2016*) are required to more fully understand how variation in growth parameters affect responses to spatial protection.

Also contrary to our predictions, interspecific differences in movement behavior did not affect responses to spatial protection by exploited species. One potential explanation is that the proportion of immature fish was greater for high mobility-benthopelagic species (95%) than for low mobility-demersal species (53%). Most mature individuals from high mobility-benthopelagic species may have already undergone ontogenetic depth shifts below the reach of our dive surveys (*Love, Yoklavich & Thorsteinson, 2002*), reinforcing the need to collect data at deeper depths to strengthen inferences. Additionally, high mobility-benthopelagic species tended to have higher $k$ values (Table 1), which might have confounded mobility effects.

The relative body sizes of high mobility-benthopelagic species increased with depth and decreased with greater topographic structural complexity, while the body sizes of low mobility-demersal species had the opposite relationship to these habitat variables. Perhaps the different responses reflect the distribution of resources being tracked (high mobility-benthopelagic species feeding more on pelagic prey near the surface: *Love, Yoklavich & Thorsteinson, 2002*) and different antipredator strategies (low mobility-demersal species relying more on crypsis and refuges in complex reefs).

The larger body sizes for exploited species within the Goose Island and West Aristazabal RCAs likely reflect reduced fishery mortality rather than environmental variation, as body sizes for the unfished Puget Sound rockfish did not differ between protected and fished

sites. Given that fecundity increases disproportionately with body size (*Dick et al., 2017*) and that larvae exports can occur from protected to fished areas (*Baetscher et al., 2019*), these RCAs may already be contributing to fishery sustainability; such benefits should increase over time (*Starr et al., 2015*; *Kaplan et al., 2019*).

At four of six RCAs that we examined, however, rockfish body sizes at protected sites were similar to or smaller than those at adjacent fished sites. There are at least three potential, non-mutually exclusive explanations that might explain this result. First, the effectiveness of a protected area generally increases with its size and age (*Molloy, McLean & Côté, 2009*). In our sample, RCA age varied little, but RCA size ranged widely. The two RCAs that included larger fish within their boundaries (relative to adjacent areas) also were the two largest RCAs that we examined (Appendix S2). While more RCAs of varying sizes need to be examined, it is plausible that RCA size influenced our results.

Second, despite analyses controlling for depth and topographic structural complexity, some of the variation in RCA performance might reflect oceanographic differences between control and protected sites (see *Caselle et al., 2015*). Within a rockfish species, growth rates are faster in oceanic areas—which have lower temperatures and higher salinity—than in inland coastal waters (*West, Helser & O'neill, 2014*). Oceanographic context, therefore, may partly explain why we found no protection benefits at the Fish Egg Inlet RCA; this RCA encompasses inland coastal waters but local geography constrained control sites to more oceanic conditions (Fig. 2), where growth rates likely are faster (*West, Helser & O'neill, 2014*). Perhaps also reflecting oceanographic factors, at the Kitasu Bay RCA a control site just outside its boundary encompassed a reef with outstanding rockfish productivity, which might explain why individuals from faster-growing species were larger outside this RCA. Similarly, the Goose Island and McMullin Group RCAs share a boundary, yet only the former had larger fish in protected than in control sites; the control sites for the McMullin Group include several locations where conditions are more oceanic than at the control sites for Goose Island, which may have influenced this result (Fig. 2).

Third, while analyses are lacking for the extent to which legal and illegal fisheries occur within RCAs of the Central Coast, the intensity of these activities might have varied between locations. Studies in southern British Columbia documented a high incidence of illegal recreational fisheries within RCAs (*Lancaster, Dearden & Ban, 2015*; *Haggarty, Martell & Shurin, 2016*). Fisher compliance tended to decline with fishing effort adjacent to the RCA and with the RCA's proximity to fishing lodges. Further, lower compliance was associated with fewer resources for enforcement and smaller RCA sizes (*Haggarty, Martell & Shurin, 2016*). Importantly, RCAs were created to protect rockfish from the highest-risk commercial and recreational fisheries, yet trap fisheries for invertebrates and mid-water trawls for groundfish remain legal within them (Appendix S1), likely causing rockfish mortalities (*DFO, 2019b*). The extent to which illegal and legal fisheries may have contributed to the lack of larger fish sizes at two thirds of the RCAs that we examined requires further investigation. Further, responses to spatial protection might be more likely to be detected if areas adjacent to reserves are fished more intensely (see *Jennings, 2000*; *Kaplan et al., 2019*), and future analyses should consider these effects.

Indigenous fisheries for traditional foods are allowed in RCAs, yet these fisheries are regulated by traditional laws derived from the long-term knowledge and stewardship principles that have allowed sustainable harvests over centuries (*Jones, Rigg & Lee, 2010*; *Ban et al., 2018*; *Ban, Wilson & Neasloss, 2020*). Hereditary chiefs remain the traditional stewards of their territories who, with technical staff, develop food fishing policies consistent with traditional laws. Such policies are operationalized via Coastal Guardian Watchmen who communicate with food fishers and patrol the waters for compliance, mitigating impacts from traditional fisheries in protected areas. Additionally, Central Coast First Nations run their own catch-monitoring programs, which they use to inform the governance and management of their traditional fisheries (*Ban et al., 2018*; *Ban, Wilson & Neasloss, 2020*). Thus, it is unlikely that Indigenous traditional fisheries contributed to the lower performance of some RCAs. Consistent with this notion, other studies in British Columbia found that the relative abundance of Dungeness crab (*Cancer magister*) increased over time after spatial fishery closures for commercial and recreational fishers where implemented, despite traditional Indigenous fisheries continuing within these closures (*Frid, McGreer & Stevenson, 2016*; *Burns et al., 2020*).

## CONCLUSIONS

At two spatial fishery closures that were only 13 to 15 years-old when most data were collected, we found that the body sizes of 12 species of exploited rockfishes were larger at protected than at fished sites. Differences between protected and fished sites were greater for species with lower values for growth parameter $k$, which was the opposite of what we expected. Also contrary to expectation, interspecific differences in movement behavior did not affect body size responses to spatial protection. Nonetheless, life history and behavioral frameworks guided the prediction and interpretation of species differences in responses to spatial protection, which is essential for the adaptive management of protected areas (*Jennings, 2000*; *Claudet et al., 2010*; *Kaplan et al., 2019*).

Canada's federal fishery management agency is reviewing the effectiveness of RCAs (*DFO, 2019a*; *DFO, 2019b*) and our results can inform this process. For four of six RCAs we found no evidence that fish were larger in protected than in adjacent fished sites. Perhaps these deficiencies in conservation effectiveness could be mitigated by increasing resources for compliance monitoring and enforcement, and potentially by modifying some RCA boundaries to increase the proportion of more productive, oceanic habitats within such boundaries.

Critically, the trophic position of rockfishes increases with their individual body sizes (*Trebilco et al., 2016*; *Olson et al., 2020*). Larger size classes of yelloweye rockfish (*S. ruberrimus*)—a long-lived demersal species inherent to Indigenous diets and undergoing size declines (*Eckert et al., 2017*; *McGreer & Frid, 2017*)—occupy a particularly high trophic position in rocky reefs (*Olson et al., 2020*). Ensuring that RCAs effectively restore large size structures for exploited species, therefore, is essential for ecosystem-based fishery management.

## ACKNOWLEDGEMENTS

Chris Rooper (CR) and Doug Neasloss provided key insights on earlier drafts. Milton Love, CR, and Dana Haggarty provided expert opinion on behavioral classes. Formal reviews from Rowan Trebilco, Scott Hamilton and an anonymous referee helped us improve the manuscript. We thank Derek VanMaanen, Kyle Hall, Andrew McCurdy, and Jarred O'Connell for their essential work during dive surveys. For other field contributions, we thank Chris Corbett, Courtney Edwards, Ernie Tallio, John Sampson and Robert Johnson. Julie Beaumont provided technical assistance in generating Fig. 1 and estimated distances to RCAs. Vernon Brown and Christina Service commented on earlier drafts. The following technical staff from First Nations communities have provided essential direction to the project: Doug Neasloss, Peter Siwallace, Jennifer Walkus, and Danielle Shaw. Data collection was observational and did not require permits.

### Funding

Research was supported by the Gordon and Betty Moore Foundation (1406.03, 2016/17), the Aboriginal Aquatic Resource and Oceans Management Program (ARM2019-MLT-1006-2), Marine Planning Partnership (P098-00591), the Aboriginal Fund for Species at Risk (2017AFSAR3044), the Oceans and Freshwater Science Contribution Program (2019-20 Partnership Fund), the Canada Nature Fund for Aquatic Species at Risk (2019-NF-PAC-001), and in-kind by the Tula Foundation. The funders had no role in study design, data collection and analysis, decision to publish, or preparation of the manuscript.

### Grant Disclosures

The following grant information was disclosed by the authors:
Gordon and Betty Moore Foundation: 1406.03, 2016/17.
Aboriginal Aquatic Resource and Oceans Management Program: ARM2019-MLT-1006-2.
Marine Planning Partnership: P098-00591.
Aboriginal Fund for Species at Risk: 2017AFSAR3044.
Oceans and Freshwater Science Contribution Program: 2019-20 Partnership Fund.
Canada Nature Fund for Aquatic Species at Risk: 2019-NF-PAC-001.
Tula Foundation.

### Competing Interests

Madeleine McGreer, Alejandro Frid, Tristan Blaine, and Hannah Kobluk are employed by the Central Coast Indigenous Resource Alliance. Sandie Hankewich and Ernest Mason are employed by Kitasoo/Xai'xais Fisheries. Mike Reid is employed by Heiltsuk Integrated Resource Management Department.

### Author Contributions

- Madeleine McGreer analyzed the data, prepared figures and/or tables, authored or reviewed drafts of the paper, and approved the final draft.

- Alejandro Frid conceived and designed the experiments, performed the experiments, analyzed the data, prepared figures and/or tables, authored or reviewed drafts of the paper, and approved the final draft.
- Tristan Blaine performed the experiments, authored or reviewed drafts of the paper, and approved the final draft.
- Sandie Hankewich and Ernest Mason conceived and designed the experiments, performed the experiments, authored or reviewed drafts of the paper, and approved the final draft.
- Mike Reid conceived and designed the experiments, authored or reviewed drafts of the paper, and approved the final draft.
- Hannah Kobluk analyzed the data, prepared figures and/or tables, and approved the final draft.

### Animal Ethics

The following information was supplied relating to ethical approvals (i.e., approving body and any reference numbers):

Data collection was observational and did not require permits from federal agencies. Further, the Wuikinuxv, Kitasoo/Xai'xais, Heiltsuk and Nuxalk First Nations hold Indigenous rights to their own territories, where all data were collected. Scientific staff involved in field research—who are members of these Nations or work directly for them—had all the approvals required by Indigenous rights holders.

### Field Study Permissions

The following information was supplied relating to field study approvals (i.e., approving body and any reference numbers):

Data collection was observational and did not require permits from federal agencies. Further, the Wuikinuxv, Kitasoo/Xai'xais, Heiltsuk and Nuxalk First Nations hold Indigenous rights to their own territories, where all data were collected. Scientific staff involved in field research—who are members of these Nations or work directly for them—had all the approvals required by Indigenous rights holders.

### Data Availability

The raw data and R code for running analyses and plotting results are available as Supplementary Files.

### Supplemental Information

Supplemental information for this article can be found online at http://dx.doi.org/10.7717/peerj.9825#supplemental-information.

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
