# Peer review of "Growth parameter *k* and location affect body size responses to spatial protection by exploited rockfishes"

_PeerJ, doi:10.7717/peerj.9825_

## Round 0.1 · original submission · Major Revisions

In general, the manuscript is well structured and presented. However, there are some issues with the writing, analysis, and reported information that should be addressed in order to improve it. See the comments given by the reviewers for more details.

·

Basic reporting

In this study, the authors examine the effects of marine protection on body size of 12 species of rockfish, inside and outside rockfish conservation areas (RCAs) closed for up to 15 years, and the effects of life history and behavioral differences on the strength of that response. The authors find that fish are larger in body size in 2 of the 6 RCAs examined and that growth parameters may explain some of the differences in the strength of the response. They also report effects of topographic complexity and depth on body size, which varies in different ways based on mobility.

The paper was short and concise and generally well written. Raw data were shared.

Experimental design

I’m not particularly in love with the length anomaly metric as the sole means of providing information on the response of fish body size to protection. I think I would like it better if this approach complimented more traditional analyses using mean lengths, comparisons of size distributions, calculations of differences in lengths in vs. out, or response ratios. I understand why the authors used this standardization approach, because it allows them to compare across species that differ greatly in body size in the same analysis, but this was not articulated clearly in the manuscript.

The methods section needs to better describe how the length anomalies were calculated and then averaged and used in subsequent analyses. Did you calculate the mean anomaly for each species in each location pooled over all years and use that in the models?

If you have sizes of each RCA, how about conducting an analysis to test explicitly whether reserve size is associated with the strength of the length response measured? The same can be done with estimates of ocean productivity or fishing pressure. With 6 RCAs, the statistical power may be low, unless the effects are strong, but it would be interesting to see if the patterns match the qualitative descriptions of the likely drivers that the authors describe in the discussion. Are there differences in habitat complexity between any of the RCAs and their control sites that may help to explain why reserve effects were detected in some locations but not in others?

Validity of the findings

The growth parameter k is not necessarily a good measure of somatic growth. Instead, it reflects how quickly the asymptotic size Lmax is attained by a species, which may or may not correspond well with somatic growth. In the VBGF model, Lmax and k are often inversely related (higher Lmax results in a smaller k). However, species with a larger Lmax often attain larger sizes and can be bigger in size at the same age compared to species with a lower Lmax and higher k. Since they are larger at the same age, this means that actual somatic growth can be higher in cases of a lower k.

Goose Island and McMullin appear to share an RCA boundary (or they are very close), yet the responses to protection are very different. Is this just an artifact of habitat and which sites are the paired control sites? Some discussion is warranted.

Table 2 includes the parameter values for the GLSM, however nowhere in the manuscript were there reports of the test statistics, significance levels and p-values for the different factors included in the model. This is much more important than the parameter values and should be what is reported, especially because there were lots of factors and interactions included in the models. Most readers care much more about the significance of the model factors than the parameters that can used to create a predictive model of the expected length anomaly at any given location (which is what Table 2 is good for).

The body size anomaly effects of mobility class and topographic complexity need to be described in more detail and interpreted more in the discussion. The results are interesting that high mobility species are large in low complexity habitats while demersal species are larger in high complexity habitats. It is also interesting how the shifts in body size with depth are different between the two mobility classes.

Show size frequency distributions of each species, but plotted by samples inside RCAs vs outside RCAs. These can be overlapping and shaded with transparency to allow the reader to see whether fish are bigger inside MPAs in general for each species. Appendix S10 shows this for each species overall, but not broken down into inside vs. outside. It would also be good to test whether the distributions differ significantly for any species or locations.

The actual data points should be shown on the plots in Fig. 3 and Fig. 4, not just the model fit and 95% CI, so the reader can see the spread of the length anomaly data as a function of growth parameter k or topographic complexity or depth in each panel. It would also be nice to see the mean anomaly +/- SE for each point (which gives different information than the overall model fit and 95% CI). Would it be possible use a different symbol for each species, so readers can see which points correspond to the different species?

Why not also include the fish density data from the surveys to test whether abundances differ inside and outside the RCAs? Or the density and size data can be combined to calculate biomass and that could be used to test for MPA effects.

Are there any effects of time on the size differences detected inside and outside MPAs? The data from the first year could be compared to data collected in the last year or two of the surveys to calculate a change in size distributions and to test whether there is evidence that the size responses have increased over time inside the RCAs relative to the fished areas.

Reviewer 2 ·

Basic reporting

Overall, the content of this manuscript is valuable to both the scientific community and ecosystem managers. The paper addresses the differences in how commercially and recreationally important fish species may differ in their response to spatial protection because of different life history strategies.

However, there are some issues with the writing, analysis, and reported information that should be addressed prior to acceptance. The introduction seemed very choppy and is not constructed in logical progression. One simple change I would recommend is to combine the first and third paragraphs. The first paragraph discusses protected areas and how they may be beneficial, the third paragraph discusses how protection benefits are measured. The fourth paragraph (line 62) is situated in the Introduction as if it was a summary of the literature, but in reality this paragraph contains a hypothesis that the authors are trying to test. The Introduction should be re-written to identify the hypothesis that “interspecific variation in somatic growth rates and the age of spatial fishery closures should influence body size responses by exploited species to spatial protection (Fig. 1).” This is an interesting hypothesis, but is not sufficiently accepted or proven to be discussed as if it was fact. With respect to Fig. 1, I would like to see the data generated by this project placed into this construct, it might lead credence to, or allow the rejection of, this unstated hypothesis. Similarly, the authors correctly identify in the Introduction that habitat type affects life history characteristics of rockfishes, but don’t clearly describe how habitat fits into their conceptual model.

Experimental design

The work presented in this manuscript is appropriate for PeerJ. The research question is relevant and meaningful and the authors show how the research addresses an identified knowledge gap. The problem I saw, however, is that the authors mixed variables together in a way that makes it difficult to address their primary hypothesis. They said they wanted to know if species life history parameters, habitats, and age of protected areas influenced body size responses (growth rates) of different species. It appears to me, however, that they lumped all fished species together, didn’t include habitats specifically, and lumped different ages of protected areas. Also, I am concerned about the use of S. emphaeus as a control fish between the protected and fished sites. That species is small and probably shows less variation in body size compared to other rockfishes. Also, I would like to see a more thorough explanation of the use of length anomalies. It is not clear why the authors chose to use length anomalies when they are trying to show that there is a difference in response of fast growing (larger) fishes vs slow growing (smaller) fishes.

One major concern I have is the statement that, “Total lengths were estimated visually with the aid of a 1-metre-long ruler attached to a pole; although these estimates have limited precision, their measurement error likely is consistent across treatments and species, not biasing results.” I would argue that this type of visual length estimate is very likely to be problematic unless the various scuba divers have trained together and evaluated their individual accuracy and precision biases. In many places around the world, researchers using underwater visual census techniques have tested their accuracy and precision by estimating the lengths of targets of known size (e.g., plastic fish). In one study I know of, observers systematically underestimated the lengths of large fishes and overestimated the lengths of small fishes, but not every observer was biased in the same direction. I would like to see the authors address this issue more carefully.

Validity of the findings

I liked the creativity and analytical approach taken by the authors. I would like to see more of the author’s work in the results section rather than in the supplementary material. A major portion of the results discuss the percentages of fish captured at an RCA since implementation. However, time since RCA implementation was not a factor in any of the analyses presented and is confounded by the factor of location.

I think conclusions are well stated along with possible factors that could be influencing the results. The authors provided good recommendations about modifying RCA’s . One question I have is that while the authors go into detail about illegal fishing activity within the RCA and how that could be affecting their results, I am curious about the amount of fishing outside of the RCA’s in your reference sites. Could some reference sites be experiencing low to moderate fishing while others are experiencing high fishing? How could that affect the differences that you are seeing, or not seeing, between the protected sites and the adjacent fishes sites.

Additional comments

I like the questions you are asking about the differences in response of fishes with different life history characteristics to protections offered by MPAs.

·

Basic reporting

This article is well written and generally well-presented. My main suggestions for improvements are:
(1) I think it is important to visualise the data (not just statistical fits) and some information presented in tables could be more usefully presented in figure form;
(2) Consider adding some nuance to the expectations for responses to exploitation/protection
(3) There are a few places where additional references could be added to better situate this work with relation to previously published work;
I expand on these 3 points and provide additional specific suggestions below. Please also refer to my comments in the "comments for the author" section for context.

1) visualising the data
- As a general principle, in studies based on primary data, in my view the data should always be visualised (ideally in a way that directly relates to the statistical models being fit) so that the reader can visually assess the patterns. Just showing the statistical fits can be a bit misleading. I acknowledge that you do present the diagnostic plots in the supplementary materials, and that interested readers can grab the data and make plots themselves, but I still think it is preferable to show the data in the main manuscript. In this case, ideally points would be added to figs 3 and 4 showing the raw data that the models are fit to.
- The information presented in table 2 could be more usefully presented in the form of a "parameter plot". Most importantly, parameter plots allow for easy comparison of the magnitude of parameter estimates and whether CIs around estimates include 0.

2) expectations for responses to exploitation/protection
- it struck me that the expectations explained from lines 62-72 and in figure 1 might be a bit of a simplification, as the expectations for responses of fast growing species in particular will be dependent upon the intensity of exploitation (along with how fast growing the species is). For the fastest-growing species, it might be expected that there would not be any appreciable impact on body size distribution until exploitation reaches a very high intensity. It would be good to add this nuance to the explanation. It would also be helpful to make it clearer in lines 62-71 that the explanation is in terms of relative (not absolute) body size. This is apparent in figure 1 and later in the main text, but not clear at lines 62-71.

3) framing with respect to the literature:
- line 59: consider adding papers that directly coniser impacts of exploitation on size structure of reef fish e.g.
Zgliczynski, B.J., Sandin, S.A., 2017. Size-structural shifts reveal intensity of exploitation in coral reef fisheries. Ecological Indicators 73, 411–421. https://doi.org/10.1016/j.ecolind.2016.09.045

Dulvy, N.K., Polunin, N.V., Mill, A.C., Graham, N.A., 2004. Size structural change in lightly exploited coral reef fish communities: evidence for weak indirect effects. Can. J. Fish. Aquat. Sci. 61, 466–475. https://doi.org/10.1139/f03-169

- line 88: consider adding papers led by indigenous authors at line 88. e.g.
Jones, R., Rigg, C., Lee, L., 2010. Haida Marine Planning: First Nations as a Partner in Marine Conservation. E&S 15, art12. https://doi.org/10.5751/ES-03225-150112

Jones, R., Rigg, C., Pinkerton, E., 2017. Strategies for assertion of conservation and local management rights: A Haida Gwaii herring story. Marine Policy 80, 154–167. https://doi.org/10.1016/j.marpol.2016.09.031

- lines 100 -- 111: It would be good to provide some context for expectations around the effects of habitat on size structure given that this features prominently in analyses. e.g.
Trebilco, R., Dulvy, N., Stewart, H., Salomon, A., 2015. The role of habitat complexity in shaping the size structure of a temperate reef fish community. Mar. Ecol. Prog. Ser. 532, 197–211. https://doi.org/10.3354/meps11330

- line 301: Trebilco et al 2016 demonstrated increasing trophic position with body size among rockfish prior to the article by Olson et al. cited here.
Trebilco, R., Dulvy, N.K., Anderson, S.C., Salomon, A.K., 2016. The paradox of inverted biomass pyramids in kelp forest fish communities. Proc. R. Soc. B 283, 20160816. https://doi.org/10.1098/rspb.2016.0816

# additional specific suggestions and corrections
Abstract
- line 26: consider rewording the sentence beginning "For 12 species" to "Here, using a visual survey data collected from inside and outside Rockfish Conservation Areas on the Central Coast of British Columbia". This could also note that these data were collected by the Central Coast Indigenous Research Alliance, in collaboration with First Nations. This is a really outstanding aspect of this work.
- line 40: consider rewording "frameworks can predict and help interpret..." to "theory provide a useful lens for framing and interpreting..."

Discussion:
- line 222: consider adding "better" with "more fully"
- line 228: add "mature" after "most"
- line 242: "exclusive" missing after "mutually"
- line 248: consider replacing "controlling for key habitat variables" with "accounting for habitat structural complexity"
- line 252--257: consider adding some mention of potential importance of algal cover and kelp canopy structure.

Tables:
- consider adding RVI (relative variable importance) values to table 2 to give some indication of how consistently variables were included in the best-supported models, without having to refer to the supplementary materials. Also see my comment above --- this would be better as a figure.

Figure 1
- consider adding "with respect to growth rate" to main title.
- add definition of k parameter to the caption
- consider making labelling on the plot axes more readily interporable (e.g. change "length anomaly" to "length anomaly (observed size relative to species average) and "lower (smaller)" and "higher (larger)" instead of just "lower" and "higher")

Figure 2
- are points not shown for site locations for a particular reason? If not, I'd suggest showing them. Open circles could still work if points overlap.

Figures 3 & 4
- Invoke Ram Myers' 3 golden rules of data analysis: Plot the data, Plot the data, Plot the data.

Throughout
- be consistent with a convention for presenting common names followed by latin names in parentheses or vice versa. At present it's a bit haphazard.

Experimental design

The experimental design is appropriate and sample sizes are impressive given the logistical challenges of data collection.

Validity of the findings

Findings are valid and conclusions are well stated, with clear links to supporting results. The statistical analysis is rigorous and intelligent. Interpretations and speculation are articulated appropriately.

Additional comments

I enjoyed reading this manuscript and think that it will be a valuable addition to the literature. It is well-conceived and well-written, the statistical analyses are rigorous and appropriate, and the authors do an exemplary job of making the analyses transparent and repeatable.

I think the manuscript is at a publishable standard in its current form (with a few minor typographical corrections), but I also offer some suggestions for improvement that I would encourage the authors to consider under 'basic reporting'.

---

## Round 0.2 · accepted · Accept

I am pleased to confirm that your paper has been accepted for publication in PeerJ.

Thank you for submitting your work to this journal.

·

Basic reporting

See comments on previous review. General comments are below.

Experimental design

See comments on previous review. General comments are below.

Validity of the findings

See comments on previous review. General comments are below.

Additional comments

The authors have generally done a good job revising the manuscript to address my comments and those of the other two reviewers. The changes to the introduction and discussion in particular have helped to streamline and improve the clarity and focus of the study. In addition, the parameter plot (new Fig. 2) makes it much clearer which factors are important in explaining size differences of fish assemblages inside and outside of MPAs. I am a bit confused though because on line 238 of the results and later in the discussion they state that “The effects of spatial protection on the body sizes of exploited species depended on growth parameter k and RCA location, but not on behavioral class (Fig. 2; Appendices S13a, S14a).” Yet, if you look at Figure 2, behavioral class appears to be a highly significant model parameter, with a positive coefficient score and 95% CI that do not cross zero. Doesn’t this mean that there are different body size responses for the high mobility-benthopelagic group and the low mobility-demersal group? If so, which group showed a stronger body size response and why?

I was also unable to view any of the supplementary information and figures that were submitted. The file appeared to be corrupted. I tried to download it multiple times and it would not open. Thus, I was unable to evaluate the relevance of the supplementary material.

I also recommend that the authors closely read through the manuscript to make sure they are not overstating their conclusions about the effects of the British Columbia MPAs on body size responses of fish. Only 2 of the 6 MPAs showed differences in body size, meaning the majority of their sites did not show a difference. I’m not sure if this is included in the supplementary, but it would be good to investigate whether the body size differences at the 2 sites showing change were present at the time the MPAs were implemented (or as near that as possible) or whether these differences appeared more recently after the MPAs had been in place for up to 15 years. Are those body size differences pre-existing or not?

Reviewer 2 ·

Basic reporting

The revised manuscript is clear and easy to read.

Experimental design

The research question is clearly articulated, the experimental design is clearly stated along with potential limitations, and methods are describe with sufficient detail.

Validity of the findings

The authors have done a good job of describing how their data fit the predictions and discuss reasons why results may have differed from expected results.

Additional comments

This manuscript is greatly improved from the version I reviewed. The authors addressed all the concerns I had.

·

Basic reporting

Please refer to general comments to authors

Experimental design

Please refer to general comments to authors

Validity of the findings

Please refer to general comments to authors

Additional comments

As I noted in my previous review, my assessment was that the original submission (prior to revisions) was of a publishable standard, but I offered several suggestions for improvement. The assessment of the other reviewers was that more substantial revisions were necessary.

I commend the authors for undertaking a comprehensive and considered revision of this article, which has been substantially strengthened. I thought their responses -- both to my comments and those of the other reviewers -- were thorough and well reasoned.

I support the publication of this manuscript in its current form.